# Localization of Sliding Movements Using Soft Tactile Sensing Systems with Three-axis Accelerometers

**DOI:** 10.3390/s19092036

**Published:** 2019-04-30

**Authors:** Hiep Xuan Trinh, Yuki Iwamoto, Van Anh Ho, Koji Shibuya

**Affiliations:** 1Department of Mechanical and Systems Engineering, Faculty of Science and Technology, Ryukoku University, 1-5 Yokotani, Seta Oe-cho, Otsu, Shiga 520-2194, Japan; t18m026@mail.ryukoku.ac.jp (Y.I.); koji@rins.ryukoku.ac.jp (K.S.); 2School of Material Science, Japan Advanced Institute of Science and Technology (JAIST), 1-1 Asahidai, Nomi, Ishikawa 923-1292, Japan; van-ho@jaist.ac.jp

**Keywords:** soft tactile sensing system, localization and object detecting, three-axis accelerometer

## Abstract

This paper presents a soft tactile sensor system for the localization of sliding movements on a large contact surface using an accelerometer. The system consists of a silicone rubber base with a chamber covered by a thin silicone skin in which a three-axis accelerometer is embedded. By pressurizing the chamber, the skin inflates, changing its sensitivity to the sliding movement on the skin’s surface. Based on the output responses of the accelerometer, the sensor system localizes the sliding motion. First, we present the idea, design, fabrication process, and the operation principle of our proposed sensor. Next, we created a numerical simulation model to investigate the dynamic changes of the accelerometer’s posture under sliding actions. Finally, experiments were conducted with various sliding conditions. By confirming the numerical simulation, dynamic analysis, and experimental results, we determined that the sensor system can detect the sliding movements, including the sliding directions, velocity, and localization of an object. We also point out the role of pressurization in the sensing system’s sensitivity under sliding movements, implying the ideal pressurization for it. We also discuss its limitations and applicability. This paper reflects our developed research in intelligent integration and soft morphological computation for soft tactile sensing systems.

## 1. Introduction

Tactile sensors, according to [1], are defined as devices that can detect and measure a contact’s properties in a predetermined area and subsequently pre-processes the signals with the sensing elements before sending them to higher levels of perceptual interpretation. Human tactile sensing generally serves as a reference point in robotics. Human sensory psychophysics can be described by four main attributes: location, modality, intensity, and timing [2]. Many researchers focus on artificial tactile sensors that mimic these capabilities for applications in robotics [3,4,5]. In terms of localization and object detection, research is currently focusing on designing and fabricating artificial tactile sensor arrays that can cover large surfaces to obtain rich contact information. For instance, Damian et al. [6] presented an artificial ridged skin for detecting the position and speed of a sliding object using force sensors that were embedded in nonuniform arranged-parallel ridges. Another work [7] proposed a real-time sensor fusion algorithm for estimation object position, translation, and rotation during grasping by pressure array sensing. Although these tactile sensor arrays can detect an object’s localization on a large contact surface with improved accuracy and spatial resolution, they require many sensing elements, complicating integration and manufacturing. 

In addition, due to the complexity of the output data, tactile sensing array normally needs a computational program with complicated algorithms for data processing [8]. 

Recent research in morphological computation and embodied intelligence is introduced as a method to reduce the complexity of entire robotics systems by creating different sensing and actuating functions [9,10]. Soft robotics is an example that utilizes embodiment to facilitate control and perception [11]. One crucial element of soft robotics research is soft tactile sensors, which are normally implemented by embedding sensing elements in a soft body [12]. The sensing elements can be such conventional sensors as strain gauges [13], magnets and Hall sensors [14], or pressure sensors [15], which transduce physical interaction into electrical output signals through the soft body’s deformation. By exploiting the soft, elastic properties of soft material, soft tactile sensors suggest the potential of using the characteristics of a biological body to reduce complexity [16]. Based on the idea of morphological computation to obtain rich sensory information for soft tactile sensors, we previously proposed Wrin’Tac, a soft active tactile sensor system that can change its sensitivity for different sensing tasks using a strain gauge as its only sensing element [17,18]. Although Wrin’Tac can successfully change its sensitivity and the sense sliding movements on its surface, it cannot localize them because the strain gauge’s output is only one dimensional, which means that we cannot detect the position of a sliding movement if the sensor provides identical output for different movements. Inspired by how a spider detects its prey, we present in this paper a tactile sensing system that is sensitive to different sliding motions in large contact areas using just a three-axis accelerometer as a sensing element. This sensing system consists of a soft base with a chamber covered by a thin silicone skin in which a three-axis accelerometer is embedded. We integrated pneumatic actuation to generate a large inflated skin’s surface that easily transforms the deformation at the contact’s position to the accelerometer’s position. Under a sliding motion, the accelerometer’s posture changes continuously, depending on the contact’s location, resulting in a variation of an accelerometer’s output voltages. Thus, our sensing system can detect and localize sliding motions with different directions. Several studies have utilized accelerometers for fabrication of tactile sensors [19,20,21]. A previous work [19] used a miniature accelerometer that was attached to a human finger to measure the acceleration at the radial skin to estimate the surface undulation. It was attached to several pneumatic grippers to classify the hardness of different cylinders, to estimate the pneumatic pressure, and to assess the firmness of eggplants and mangoes [20]. The authors of [21] proposed a bio-mimetic fingertip on which they embedded three commercial accelerometers and force sensors to detect force and vibration modalities. Most accelerometer applications in tactile sensors focus on estimating physical quantities such as hardness and firmness and using measured vibrations for surface identification. In terms of detecting and localization sliding movements, there is a lack of interest in accelerometer applications in tactile sensing systems. 

In this paper, to clarify this sensing system’s ability, we constructed a numerical analysis model by 3D simulation and conducted experimental validations with dynamic analyses in different sliding actions. Our proposed sensing system has the following advances: easy fabrication, low cost, and simple integration to soft robotics. Our research is using intelligent embodiment to reduce the complexity of soft tactile sensor systems. The following are the main contributions of this paper:(1)We integrated our proposed idea, design, and fabrication of a soft tactile sensing system a pneumatic actuator and a three-axis accelerometer as a sensing element for detecting and localization sliding movements in a large contact area.(2)We clarified the operation principle of our sensing system and conducted a numerical simulation to study the dynamic responses of the sensing elements under a sliding motion with morphological changes.(3)We proposed dynamic analyses and interpolated functions based on experiment results to verify the sensing system’s ability.(4)We elaborated the role of pressurization on the response of the sensing system under sliding movements.

## 2. Materials and Methods

### 2.1. Idea 

Nature provides many examples of amazing touch organs with tactile sensing that have inspired robotic applications [22,23,24]. Our study was inspired by the prey detection of spiders, which lurk at the bottom of their webs. Although spiders do not have great eyesight, and they usually use the vibrations of the web strands to locate their prey. Spiders interpret vibrations with organs on their legs called slit sensillae, which are small, hypersensitive grooves that deform with even the slightest disturbance [25,26]. 

Based on the notion of prey detection by spiders, we propose an active tactile sensing system for detecting and localizing sliding objects on large contact areas. Our tactile sensing system integrates a sensing element that is embedded in soft skin and actuation that can generate a curved skin’s surface on a sensing system to resemble the role of spider’s web (Figure 1b). We deploy a three-axis accelerometer as the sensing element and pressurized air as the actuation. Under pressurization, a curved surface was generated, and when an object contacts and slides on it, the deformation at the contact’s position is conveyed to the accelerometer’s location that provides information about the acceleration of the three axes (x, y and z axes). This information is sufficient for detecting the sliding directions and localizing the sliding movement. In our sensing system, the inflated skin’s surface spreads the deformation, which mimics the role of a spider’s web in a vibration’s transmission. Figure 1 shows schematic drawings of our soft tactile sensing system.

In this paper, we focus on the localization of sliding movements in two X- and Y- directions with different distances from a sliding line to the accelerometer (Figure 1b). For simplicity, we suppose that first a sliding object slightly contacts the soft skin (outside the inflated area) and then slides over the inflated area at a constant velocity without changing the initial contact depth. The localization of the sliding object can be characterized by the sliding directions and its position coordinate (x,y). For the X-sliding case, y is the distance from the sliding line to the accelerometer and x=vt whereas for the Y-sliding case, x is the distance from the sliding line to the accelerometer and y=vt. Here v is the magnitude of sliding velocity. Thus, for localization, first we demonstrate that the sensing system can detect the sliding motion and its directions. Based on the accelerometer output, the sliding velocity’s magnitudes and the distances from the sliding line to the accelerometer are estimated. Then the sliding object can be localized.

### 2.2. Design and Fabrication

Our tactile sensor system has two parts. The first is a soft, 64 × 64 × 20 mm base with a 50 × 50 × 10 mm chamber is cut from its surface. The soft base was made from KE-1603 (Shin-Etsu, Tokyo, Japan) silicone rubber from Shin-Etsu Silicone [27] and fabricated with a 3D printing mold. Then the silicone rubber in a liquid state was poured into it and cured. The second part is soft, thin 60 × 60 × 4 mm skin made from EcoFlex10 (Smooth-on, Macungie, PA, USA) that completely covers the chamber’s base.

To fabricate the skin, we used a small chamber to install the accelerometer using a mold. After installing an accelerometer in the small chamber, silicone rubber in liquid state EcoFlex10 part A was mixed with EcoFlex10 part B [28] at a ratio of 1:1 with a mixer for three minutes, and then the mixed liquid was poured to entirely cover the skin’s surface and cured at 70 °C for 40 min. The 3 × 5 × 1 mm accelerometer used in this paper is MMA7361L by Freescale Semiconductor (Austin, TX, USA) mounted on a 10 × 10 × 1.7 mm circuit board [29]. Next the soft skin with an embedded accelerometer was attached to the soft base with adhesive glue that is specialized for silicone rubber attachment. In the final step, a 4-mm diameter tube was put and glued for directing the compressed air in the chamber. The design and a photograph of the tactile sensing system are shown in Figure 2.

### 2.3. Operation Principle

The sensing element used in this research is a MMA7361L, a surface-micromachined integrated-circuit accelerometer by a Freescale Semiconductor, which can provide three-axis acceleration information. The device consists of a surface-micromachined capacitive sensing cell and a signal conditioning ASIC contained in a single package. ASIC uses switched capacitor techniques to measure the capacitors and provide a high-level output voltage that is ratiometric and proportional to the acceleration [29].

In our research, we use the ability to calculate the static acceleration based on the posture of the accelerometer. When the object contacts the inflated soft skin surface, the contact area is deformed. Due to the soft properties of the skin material (Ecoflex10) the skin area at the sensing element’s position is also deformed, changing the accelerometer’s posture. When the accelerometer’s posture changes, its *xyz* axis coordinate also rotates (Figure 3a,b). With each position of the accelerometer, its output is the value of the projection of the gravitational acceleration on the Ox, Oy, Oz axis.

Let θ be the tilt angle of each axis with horizontal axis Δ (Figure 3c,d) where the positive direction of θ is clockwise (the rotational direction toward gravitational acceleration g). That means according to the smallest angle to coincide with *x*, *y* or *z* axis, the value of θ is positive if the rotational direction of horizontal axis Δ is clockwise and if θ has a negative value in the contrast rotational case.

Figure 3 shows that for each axis the acceleration is:(1)a=gsinθ.

For the Ox, Oy, Oz axis, output acceleration $a_{x},a_{y},a_{z}$ can be calculated:(2)$$a_{x}=gsin\theta_{x},\;a_{y}=gsin\theta_{y},\;a_{z}=gsin\theta_{z}\;.$$

Output voltage $V_{x},\;V_{y},\;V_{z}$ of the accelerometer is:(3)$$V_{x}=ka_{x},\;V_{y}=ka_{y},V_{z}=ka_{z},$$
where k is the calibration factor.

When the object slides on the sensor’s surface, the accelerometer’s posture changes continuously, depending on the contact position. Angles $\theta_{x},\theta_{y},\theta_{z}$ are functions of time.

Finally, we get:(4)$$V_{x}=ka_{x}=kgsin\theta_{x}\left(t\right)=V_{x}\left(t\right)\;V_{y}=ka_{y}=kgsin\theta_{y}\left(t\right)=V_{y}\left(t\right)\;V_{z}=ka_{z}=kgsin\theta_{z}\left(t\right)=V_{z}\left(t\right).$$

The values of output voltages $V_{x},\;V_{y},\;V_{z}$ depend on angles $\theta_{x}\left(t\right),\;\theta_{y}\left(t\right),\;\theta_{z}\left(t\right).\;$Thus, output voltages $V_{x},\;V_{y},\;V_{z}$ depend on the contact’s position of the sliding object on the sensor’s surface. Based on the values of $V_{x},\;V_{y},\;V_{z}$, the sensing system can detect and locate the sliding movements.

## 3. Numerical Simulation

As mentioned in Section 2.3, when the object slides on the inflated skin’s surface, the accelerometer’s posture and its coordinates change, depending on the contact’s localization. Therefore, to clarify the effect of the sliding action on the accelerometer’s posture, first we exploited a finite element (FE) model with commercial software (Abaqus, Dassault Systemes, Johnston, IA, USA) to conduct a dynamic simulation.

Compared to the design of the soft tactile sensor system in Figure 1, we proposed a three-dimensional (3D) model (Figure 4) that is comprised of four different parts: a soft base, a skin layer, an accelerometer inside the skin, and a hemi-spherical indenter. The sizes of the soft base, the width of the skin layer, and the accelerometer were identical to those mentioned in the sensor system’s design. To reduce the computational cost, the thickness of the skin layer and the accelerometer were set smaller than their values in the design: 2-mm skin layer and 0.5-mm accelerometer in simulation compared to 4 mm and 1 mm in the sensing’s design. The hemi-spherical indenter has a diameter of 6 mm. The values of the Young’s modulus of the soft base and the skin were set to 1.2 and 0.07 MPa. The indenter and the accelerometer are expected to be rigid with a much larger Young’s modulus (2000 MPa) than the skin and the base. The FE model is meshed with 47,656 linear hexahedral elements of type C3D8R.

Each simulation trial was conducted in four continuous steps: pressurization of the chamber, push indenter, forward and backward slides of the indenter. In the first step, the chamber is pressurized from 0 Pa to a particular value in 0.5 s. In the second step, the indenter is displaced 0.5 mm toward the skin for 0.25 s. The interaction between the indenter and the skin surface is modeled as normal and tangential contact using the penalty formulation with a friction coefficient of 0.1. In the third step, the indenter is forced to slide horizontally right to left (forward direction) at a constant velocity of 200 mm/s in 0.25 s without changing the contact depth. In the last step, the indenter is slid back at the same velocity to its original position (backward direction). The third and last steps were conducted in both the X- and Y-sliding directions (Figure 4). Some simulation trials were also conducted with different pressurizations for comparison.

In the simulation, the accelerometer was attached to local coordinate $O_{1}\mathrm{xyz}$, which is identical to the defined coordinate in the actual sensor system. In the $O_{1}x$, $O_{1}y$ axes, nodes 1 and 2 were chosen to represent each axis ($O_{1}x$
≡
$O_{1}\mathrm{node}\;1$, $O_{1}y$
≡
$O_{1}\mathrm{node}\;2$) (notation ≡ indicates that they coincide). In our tactile sensing system, we used output voltages $V_{x},\;V_{y}$ for detecting the object’s localization. Thus, in the simulation, we only consider the position of two axes: $O_{1}x$ and $O_{1}y$.

As discussed in Section 2.3, the accelerometer output depends on the angle between $O_{1}x$, $O_{1}y$ axes and horizontal axis. Thus, to investigate the position of the $O_{1}x$, $O_{1}y$ axes when the accelerometer’s posture changes under sliding motions, we extracted the displacements of two nodes: nodes 1 and 2.

In the xy coordinate (Figure 5), angles $\theta_{x},\;\theta_{y}$ between $O_{1}x$, $O_{1}y$ and horizontal axis Δ depends on displacement Dx,Dy of nodes 1 and 2.

For node 1, $\theta_{x}>0$ if the displacement $Dy_{node1}>0$ and $\theta_{x}<0$ if $Dy_{node1}<0$ (Figure 5a).

For node 2, $\theta_{y}>0$ if $Dx_{node2}>0$, and $\theta_{y}<0$ if $Dx_{node2}<0$ (Figure 5b).

Thus, based on the change of displacement Dy, Dx of nodes 1 and 2, we can predict the dynamic output of our tactile sensing system.

The simulation results in Figure 4 show that with pressurization, the skin layer was inflated, causing an inclination of the accelerometer’s posture, which is possibly more sensitive to sliding motions. From the simulation results in Figure 6, when the chamber is pressurized, under the indenter’s sliding traction, coordinate axis $\;O_{1}x,{\;O}_{1}y$ of the accelerometer changes correspondingly. With the X-sliding action, displacement Dy (blue line) changes from positive to negative in the forward case and negative to positive in the backward case. The value of Dx (red line) is negative for all the times in both cases. With Y-sliding, the sign of Dx changes but Dy has a constant sign. That means with X-sliding, the sign of $\theta_{x}$ changes and $\theta_{y}$ has a constant sign, and with Y-sliding, the sign of $\theta_{x}$ is constant and $\theta_{y}$ has a changed sign.

The simulation results imply that, by investigating the output acceleration of the accelemeter on the x, y axis, the tactile sensing system can detect the sliding action with forward and backward actions in both the X- sliding and Y- sliding cases.

To clarify the role of the inflated skin’s morphology, we conducted simulations with various pressurizations (Figure 6), with higher pressure values, the Dx and Dy displacement increased, indicating that the sensitivity and output voltage of the sensing system was increased with higher pressurization.

## 4. Experimental Setup

We conducted experiments with our soft tactile system to assess the possibility of localizing sliding movements on the skin’s surface. Figure 7 shows the experimental setup that consists of two linear stages (PG615-R05 AG-C, Suruga Seiki, Japan) and an indenter. The linear stages provide precise movements with 2-μm resolution. The indenter, which is a rigid spherical 6-mm-diameter ball, is attached vertically to the linear stage. The tactile sensor system was placed in a solid box fabricated by a 3D printer. The box was fixed and placed horizontally to the linear stage, which moves the sensing system back and forth. A laser sensor (Keyence-IL-030) measured the height of the inflated skin, and this value estimates the skin’s deformation and the accelerometer’s posture when the sensor system is not being probed.

We defined the two sliding directions as the X- and Y-axis directions along the x and y axes of the accelerometer. When the indenter moves toward the right relative to the sensor system, we call the direction forward and the opposite direction is backward. With both the X- and Y-sliding directions, there were six sliding cases (Figure 8a).

To investigate the sensor system’s capability to detect and localize a sliding object on a large contact area, we conducted experiments with different distances from the accelerometer to the indenter (Figure 8b).

-With X-sliding case: Distance value y increased from 0 to 25 mm or from −25 to 0 mm with increased 2.5-mm steps.-With Y-sliding case: Distance value x rose from 0 to 25 mm with increased 2.5-mm steps. At a particular distance, the sliding motions were conducted with various velocities from 0.5 to 4 mm/s with increased steps of 0.5 mm/s.

All the signals from the accelerometer were sent to ADC (24 bit, Kyowa Electronic Instruments Co., Tokyo, Japan) before being sent to a computer where the data were recorded by commercial software. The ADC converter’s sampling rate was set to 1000 Hz. The skin was inflated by pressurization with a pneumatic actuation system with a compressor (JUNair) and a regulator. The air pressure was regulated within a range of 0.01 to 0.06 MPa.

## 5. Results and Discussion

### 5.1. Output Voltage of Accelerometer Under Static Pressurization Without Sliding Motion

As discussion in Section 2.3 and the simulation results in Section 3, the skin layer was inflated under pressurization, resulting in an inclination of the accelerometer’s posture. Thus, the accelerometer’s output voltage has initial values that are caused by static pressure at the no-sliding cases. To investigate the output response of the sensor system just under the effect of sliding traction, this initial output voltage of the accelerometer was measured for each value of pressurization and subtracted from the acquired output of the accelerometer during sliding operations. In addition, based on the accelerometer’s output, we also estimated the height of the inflated skin and compared it to the measured height from the laser sensor.

Without pressurization, since the skin is flat, the accelerometer lies horizontally (Figure 9a). Under pressurization, the accelerometer inclined, and the angle between the accelerometer and horizontal plane was β . $\theta_{x}$, $\;\theta_{y}$ is the angle between axis $O_{1}x,{\;O}_{1}y$ and horizontal axis Δ (Figure 9b). Output voltages $V_{x}$, $V_{y}$ are shown in Figure 9c.

As in Section 2.3, output voltage $V_{x}$ can be calculated:(5)$$V_{x}=ka_{x}=kgsin\theta_{x}.$$

In Figure 9b, since $\beta +\theta_{x}=\pi$, $sin\theta_{x}=sin\;\beta$, we get the following:(6)$$\beta =acsin\left(\frac{V_{x}}{gk}\right).$$

The posture of the accelerometer can be evaluated from value β.

For an estimation of the shape of the inflated skin from output voltage $V_{x}$ of the accelerometer, the skin’s surface shape can be approximated by the following quadratic curve (Figure 10a):(7)$$y=-ax^{2}+h,\;\;\;\;\left(a>0\right),$$
where h is the skin’s highest height. The absolute value of the slope of the curve at $x_{0}$ is $2ax_{0}$, where $x_{0}$ represents the accelerometer’s position. The slope equals  tanβ, and we obtain
(8)$$a=\frac{\tan\beta }{2x_{0}}.$$

The distance from the y axis to the position at which the curve intersects the x axis is $\sqrt{h/a}$. If this distance is represented by *D*, the height of the inflated skin is computed:(9)$$h=\frac{D^{2}\tan\beta }{2x_{0}}.$$

Since the dimension of the skin is 50 × 50 × 4 mm, *D* = 25 mm. β is calculated from Equation (6). The computed value of the inflated skin’s height is then compared to the measured values using a laser sensor (Figure 10b) and plotted in Figure 11 with error less than 12%. Thus, based on the accelerometer’s output voltage we can accurately estimate the accelerometer’s posture as well as the inflated skin’s shape.

### 5.2. Output Voltages of Accelerometer Under Sliding Action

#### 5.2.1. Dynamic Analysis for Detecting Sliding Directions

To localize the sliding movements, first the sensing system needs to recognize the motion’s directions. In this section, we propose dynamic analysis based on the accelerometer’s output for detecting the sliding directions. We conducted experiments with various sliding tractions (Figure 8a) and divided them into three main cases (Figure 8b):(1)X-sliding case 1: Indenter slides along the X axis with contact position y < 0.(2)X-sliding case 2: Indenter slides along the X axis with contact position y > 0.(3)Y-sliding: Indenter slides along the Y axis.

When the indenter moves on the sensing system’s surface, the accelerometer’s output and its posture depend on the contact’s position. For analysing the accelerometer’s posture and its output, we divided the contact surface into four contact zones (Figure 12a).

When the indenter contacts the sensing system’s surface in contact zone 1, the accelerometer tends to incline toward point 1 (Figure 12b), and its posture with the corresponding xy coordinate is illustrated as in Figure 13a. Similarly, with contact zones 2, 3, and 4, the accelerometer inclines toward points 2, 3, and 4, and Figure 13b–d show the accelerometer’s posture and its xy coordinate, respectively. The analysis shown in Figure 13 suggests the following:-With contact zone 1: The values of angles $\theta_{x},\;\theta_{y}$ between O1x and O1y and horizontal axis Δ are negative, and thus output voltages $V_{x}$ and $V_{y}$ are negative: $\theta_{x}<0,\;\theta_{y}<0\to V_{x}<0,\;V_{y}<0$.-With contact zone 2: $\theta_{x}>0,\;\theta_{y}<0\to V_{x}>0,\;V_{y}<0$.-With contact zone 3: $\theta_{x}<0,\;\theta_{y}>0\to V_{x}<0,\;V_{y}>0$.-With contact zone 4: $\theta_{x}>0,\;\theta_{y}>0\to V_{x}>0,\;V_{y}>0$.

The output voltages of the sensing system with each sliding case are shown in Figure 14. With X-sliding case 1, when the indenter starts to make contact and slides over the inflated skin’s surface in contact zone 1, measured output voltages $V_{x}$, $V_{y}$ are negative, by keeping moving, the indenter slides to the other side of the skin’s surface in contact zone 2, output $V_{x}$ increases to a positive peak value before it returns to the initial state when the indenter moves off of the skin’s surface in contact zone 2. While $V_{y}$ increases from a negative value to an initial value. When the indenter slides backward, its contact with the skin’s surface changes inversely from contact zone 2 to contact zone 1, and thus $V_{x}$ changes from a positive to a negative value and $V_{y}$ keeps its negative value (Figure 14a).

With X-sliding case 2, the contact’s position changes from zone 3 to 4 with forward sliding and from zone 4 to 3 with backward sliding. The changed behaviors of $V_{x}$ and $V_{y}$ can be analysed as in the X-sliding case 1. $V_{x}$ increases from negative to positive, and $V_{y}\;$remains positive for forward sliding. With backward sliding, $V_{x}\;$decreases from positive to negative, and the value of $V_{y}$ remains positive (Figure 14b).

In terms of the Y-sliding case, with the forward sliding direction, the contact zone changes from 1 to 3 and $V_{y}$ < 0 rises to $V_{y}$ > 0. With backward sliding, $V_{y}$ changes from positive to negative. The value of $V_{x}$ is negative in both the forward and backward directions (Figure 14c). In other words, with X-sliding, the sign of $V_{x}$ changes but $V_{y}$ has a constant sign. Whereas with Y-sliding, the sign of $V_{x}$ is constant and $V_{y}$ has a changed sign. This resemble the discussion of the displacement’s change (Dx, Dy) in the simulation results, plotted in Figure 6, Section 3. A dynamic analysis of the experimental results agrees well with the simulation results. Based on output voltages $V_{x}$, $V_{y}$ of the accelerometer, we conclude that our sensing system can temporally detect the sliding action with two forward and backward directions in the X- and Y-sliding cases.

#### 5.2.2. Estimation of Sliding’s Localization.

We presented the idea of sliding movement localization in Section 2.1 and in Section 5.2.1 demonstrated that the sensing system can temporally detect a sliding motion and its directions. Based on the accelerometer’s output voltage, in this section we confirm that the sliding object can be localized through the distance from the sliding line to the accelerometer and the magnitude of the sliding velocity.

Because the deformation of the soft skin at the accelerometer’s position depends on the distance from the object to the accelerometer, the accelerometer’s outputs are a function of the object’s contact point. Therefore, based on the accelerometer’s output voltages, a soft tactile sensing system can estimate distances x,y between the accelerometer and the object’s sliding line (Figure 8b). To clarify this capacity, we experimented with various distances x,y for both the X- and Y-sliding cases (Figure 8b).

Figure 15a and Figure 16a show the accelerometer’s output voltages $V_{x}$, $V_{y}$ with different distances y,x of the X- and Y-sliding cases, when the soft skin is inflated by pressurization P=0.035 MPa. These figures suggest that the values of $V_{x}$, $V_{y}$ generally depend on distances y and x. When distances y,x increase, the output voltage decreases.

In Figure 15a, $V_{x}min$ and $V_{x}max$ are the minimum and maximum values of output voltage $V_{x}$ with corresponding moments $t_{min}$ and $t_{max}$. In Figure 16a, ($V_{y}min$, $t_{min}$) and ($V_{y}max$, $t_{max}$) are similarly minimum and maximum magnitudes with corresponding moments of $V_{y}$.

Figure 15a and Figure 16a show that $V_{x}min$, $V_{x}max$ and $V_{y}min$, $V_{y}max$ depend on the y and x values. Thus, based on the magnitude of the minimum and maximum voltage, the distance from the accelerometer to the object’s sliding line can be estimated and expressed as the following interpolated functions:(10)$$y=f_{1}\left(V_{x}min\right);\;y=f_{2}\left(V_{x}max\right)\;\mathrm{for}\;X\text{-}\mathrm{sliding}\;x=f_{3}\left(V_{y}min\right);\;x=f_{4}\left(V_{y}max\right)\;\mathrm{for}\;Y\text{-}\mathrm{sliding}.$$

Figure 15a and Figure 16a also indicate that the $t_{min}$ and $t_{max}$ values are not dependent on distances y or x. $t_{min}$, $t_{max}$ only depend on the magnitude of velocity v (Figure 15b and Figure 16b).

The magnitude of the sliding velocity can be expressed by:(11)$$v=\chi_{1}\left(t_{min}\right);\;v=\chi_{2}\left(t_{max}\right)$$

From our experimental results with various values of distances and velocity magnitudes, we extracted the values of the minimum, maximum output voltage, and corresponding moment $t_{min},t_{max}$ to assess the interpolated functions (Figure 17, Figure 18, Figure 19 and Figure 20). These functions estimate the contact’s distances x or y and velocity’s magnitude v. The dependences of the velocity’s magnitude on the values of $t_{min},t_{max}$ and distances y, x on $V_{x\;min},\;V_{y\;min}$ are not linear. Thus, we chose quadratic equations and cubic functions for the interpolation of the velocities and the distances. These functions create the best fit experimental data without requiring complicated computations.

Based on the output voltages of the accelerometer with the interpolated functions, our tactile sensing system can estimate the magnitude of the velocity and the distance from the accelerometer to sliding lines x or y. Figure 21 and Figure 22 present the calculated and actual values of the velocity and the distance from the accelerometer to the object’s sliding line with an error bar.

To estimate the velocity, the maximum errors between the calculated and actual magnitude were less than 8%. This indicates the sensing system’s accuracy for the velocity’s estimation. Note that when the velocity’s magnitude increases, the values of $t_{min},t_{max}$ decrease, and thus based on experiments we acknowledged the sensing system can estimate a maximum velocity of about 13 mm/s.

To estimate the distance from the accelerometer to the object’s sliding line, for the X-sliding direction, Figure 21a shows that the maximum error is less than 15% with distance y≥5 mm. Nonetheless, with distance y=2.5 mm, the error is about 40% due to the effect of the nonlinearity of the large deformation at the accelerometer’s position when the indenter is close to the accelerometer. The experimental results presented in Figure 15a and Figure 16a also show that the output voltage decreases when the distance increases. If the minimum and maximum output voltages approach zero, the sensing system cannot detect the sliding actions. Therefore, for the X-sliding case, the tactile sensing system has a detectable area with distance −17.5≤y≤17.5 mm (Figure 23a), and in this area the sensing system can detect the sliding direction and estimate the velocity’s magnitude where the maximum error is less than 8%.

Inside the detectable area, there are two effective localization areas with distances −17.5≤y≤−5 mm and 5≤y≤17.5 mm (blue area in Figure 23a), where the sensing system can estimate the distance from the accelerometer to the object’s sliding line where the maximum error is less than 15%. Similarly, in the Y-sliding direction, the detectable area is evaluated with distance 0≤x≤25 mm and the effective localization area is 10≤x≤22.5 mm (Figure 23b).

In the effective localization area, based on the interpolated values of velocity magnitude v and the distance between the accelerometer and the sliding line, the sliding object can be localized through the values of its coordinates x and y. For the X-sliding case, (x,y)=(vt, y), and for the Y-sliding case, (x,y)=(x, vt). Figure 24 and Figure 25 illustrate the comparisons of the actual object’s position and its estimated localization with various sliding cases.

Because we use interpolated functions to estimate the values of sliding velocity v and distances x,y, for the sliding object’s localization, the sensing system needs a delayed period for the accelerometer’s output voltage to reach extreme values. The value of the delayed time depends on the magnitude’s velocity and can be estimated from interpolated function $v=\chi_{1}\left(t_{min}\right);\;v=\chi_{2}\left(t_{max}\right).$

In the X-sliding case with forward direction, the delayed time is about $t_{delay}=\frac{18}{v}$ (s), and with backward direction, it is $t_{delay}=\frac{20.5}{v}$ (s). For the Y-sliding case with forward direction, $t_{delay}=\frac{17.5}{v}$ (s) and with backward direction: $t_{delay}=\frac{21.5}{v}$ (s), compared to the total time of one sliding action on the inflated skin’s surface $t_{sliding}=\frac{50}{v}$ (s). This result denotes that in a sliding action, after the indenter slides about $\frac{2}{5}$ of the distance, the sensing system can localize its position with the maximum errors are less than 15%.

### 5.3. Role of Pressurization

In this proposed tactile sensing system, inflated soft skin is generated by the activation of pressurization. Pressurization plays the actuation role that creates the continued change in the accelerometer’s position under the object’s sliding action. Figure 11 in Section 5.1 shows that with higher pressurization, the inflated skin’s height increases, leading to higher deformation of the skin’s surface under identical sliding motions. Resulting in increased output voltage of the accelerometer. Thus, the pressure’s value greatly affects the sensitivity of the sensing system with sliding motions.

Figure 26 shows the output voltage responses of the accelerometer under sliding actions with various pressurization’s conditions. With higher pressure value, the output voltage increases. Thus, sensitivity to the sliding action increases with extra pressurization. This result resembles the simulation results in Figure 6 of Section 3 and suggests the potential of numerical simulations for predicting the responses of our sensing system under various sliding actions, although in the numerical simulations, the pressurization was setup at a much smaller value than in experiments to save computational cost. Thus, by confirming the simulation and experimental results, we conclude that with higher pressure values, the sensing system’s sensitivity increases with sliding motions.

Nonetheless, at higher pressure (P=0.045 MPa) the output voltage is not much bigger than at pressure P=0.035 MPa due to the nonlinearity of the skin’s deformation with high pressure. In addition, high pressurization might prevent sliding movements. With the presented configuration of the soft active tactile sensor system in this paper, the ideal pressurization is 0.035 MPa, which was used in the experimental results in Section 5.2. This result also suggests that for generating and maintaining the inflated skin’s surface, the required pressurization is small (maximum, 0.035 MPa).

### 5.4. Limitations and Applicability of Sensing System

We analyzed our sensing system’s ability on the detecting and localization of sliding movements and validated it by simulations and experiments. Nonetheless, it continues to suffer from fundamental issues and limitations.

The accelerometer’s output depends on the contact location and the contact force. We assume that first the sliding object slightly contacts the soft skin outside the inflated area and slides without changing the initial contact depth. Thus, under specific pressurization, the contact force at each contact point is identical for all sliding movements. We can ignore the effect of the contact force on the accelerometer’s output voltage. In the case of changing the contact depth during sliding movements, the effect of the force at each contact point must be considered. To solve this issue, in the next step we will propose a finite element model to estimate the deformation of the inflated soft skin’s surface under various contact forces. This model could also be used to locate the indentations on soft skin and estimate the normal contact force. In addition, the effect of acceleration due to the friction force between the sliding object and the skin’s surface was not mentioned in this paper. This effect could be reduced by filtering, such as with a Kalman filter.

A sensing system can be used for specific devices and an accelerometer’s output may change if the entire device changes its posture. Thus, this effect must be investigated and eliminated. To solve this issue, it may require an algorithm that calculates the accelerometer’s output based on the whole device’s posture. These calculated values will be subtracted from the accelerometer’s acquired output for sensing.

Another current limitation of our sensing system is that it can detect and localize sliding motions with two X- and Y- directions because we only deploy the accelerometer as a sensing element. In the future we will use more accelerometers, each of which will be placed at a suitable initial posture and position. Based on the output responses of all the accelerometers, we will build a method using machine learning to localize the sliding motions with diagonal directions. To accomplish complicated tasks, a sensing system must be trained with training data in various conditions of sliding motions. After training, the acquired program can be used as the sensing system’s brain that should be attached to the main processing system. The processing system can help a sensing system be localized temporally with improved revolutions. We must tackle this critical issue in the future to fulfill this sensing system. In addition, we will use another accelerometer with a flexible circuit to replace the current accelerometer with a solid circuit board. For example, the previously proposed flexible one [21] would be easier to embed in soft skin.

Our sensing system is anticipated for applications in soft robotics. For instance, it could be attached to a soft robotics hand’s palm or other body parts of robots that frequently make contact with sliding objects. Normally, without pressurization, a soft skin surface plays the role of robotic skin and is inflated for detecting and localization sliding tasks, which are crucial for robots to make subsequent decisions. Because the pressurization for activating the inflated skin is small (maximum, 0.035 MPa), a sensing system can be activated with a small pump and a simple pneumatic controller without affecting the portability of the whole device. In addition, our sensor system is a promising device for detecting sliding objects and localizing their position in an environment with limited vision. In this paper we avoided applying our sensing system for specific applications. Instead, we set a prototype design, fabrication, and experiment with analysis to validate its ability. Thus, other researchers can exploit it to design specifically purposed structures.

## 6. Conclusions

We introduce a soft tactile sensing system with a three-axis accelerometer for detecting and localizing sliding motions on large contact surfaces. Our numerical simulation and experimental results show that it can detect temporally sliding directions in both X- and Y-direction sliding cases. In addition, interpolated functions were given to estimate the magnitude’s sliding velocity and the distance from the accelerometer to the sliding line to localize the sliding object. The detection area also validated that the sensing system can detect and localize sliding tractions on large contact areas. Future work will investigate the sensing system’s ability with various sliding cases, including diagonal directions and changing the contact depth by proposing finite element model and machine learning methods. Then we will apply our sensing system to soft robotics.

## Figures and Tables

**Figure 1 sensors-19-02036-f001:**
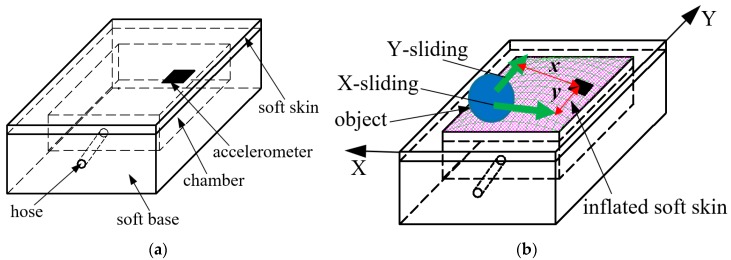
Schematic drawing of soft tactile sensing system: One soft skin layer with embedded accelerometer covers another soft base. In this base, a chamber is cut from its surface and has a hose for input compressed air. Under pressurization inside it, soft skin is inflated, changing the accelerometer’s posture. When object slides on inflated soft skin, accelerometer is sensitive to deformation at contact positions. (**a**) Initial state: without pressurization, soft skin is flat. (**b**) With pressurization, soft skin is inflated, making sensitivity of soft tactile system to sliding action on its surface.

**Figure 2 sensors-19-02036-f002:**
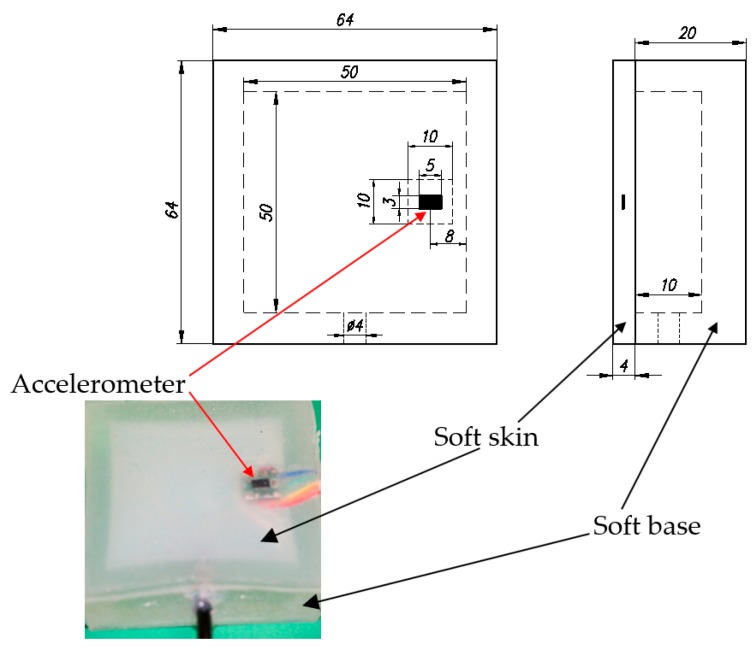
Dimension and photograph of fabricated tactile sensing system.

**Figure 3 sensors-19-02036-f003:**
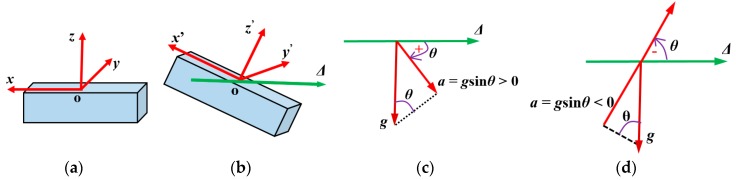
Schematic diagram shows change of accelerometer’s posture, its coordinates, and calculation of acceleration on each axis based on gravitational acceleration and tilt angle. (**a**) Initial position of accelerometer and its coordinate Oxyz. (**b**) Coordinate Ox’y’z’ when accelerometer’s posture changes. (**c**) Acceleration on axis when tilt angle θ>0. (**d**) Acceleration on axis when tilt angle θ<0.

**Figure 4 sensors-19-02036-f004:**
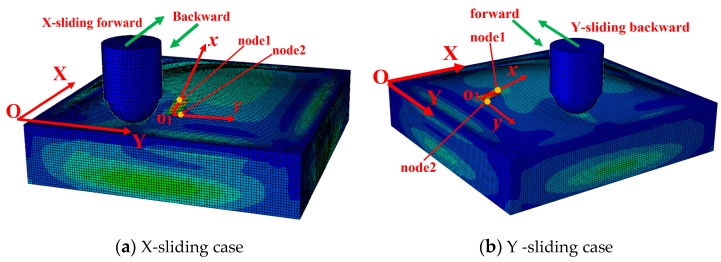
Dynamic simulation of soft active tactile sensing system. (**a**) Indenter is displaced 0.5 mm toward the skin, then is forced to slide along X axis with forward and backward directions. (**b**) Indenter is displaced 0.5 mm toward the skin, then is forced to slide along Y axis with forward and backward directions.

**Figure 5 sensors-19-02036-f005:**
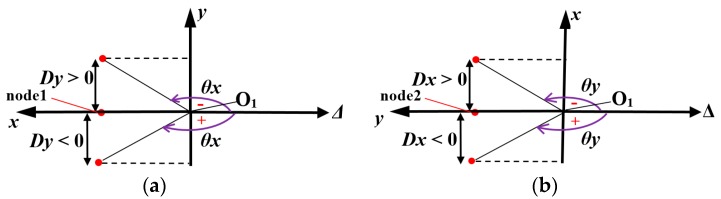
Relationship beween displacement of node 1 (Dy), node 2 (Dx) with value of angle $\theta_{x},\;\theta_{y}.$ (**a**) Displacement of node 1 Dy represents position of axis $O_{1}x$ and value of angle $\theta_{x}$. $Dy>0\to \theta_{x}<0,\;\;Dy<0\to \theta_{x}>0$. (**b**) Displacement of node 2 Dx represents position of axis $O_{1}y$ and value of angle $\theta_{y}.\;Dx>0\to \theta_{x}<0,Dx<0\to \theta_{x}>0$.

**Figure 6 sensors-19-02036-f006:**
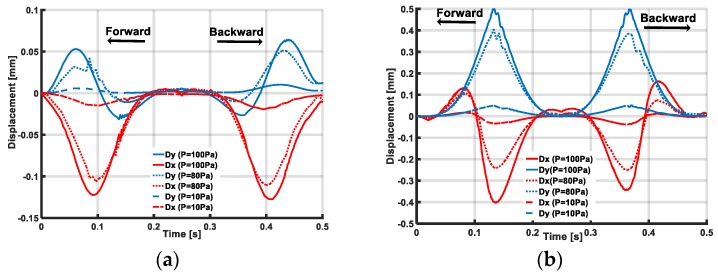
Simulation results show displacements Dy, Dx under various pressurizations: (**a**) X-sliding and (**b**) Y-sliding.

**Figure 7 sensors-19-02036-f007:**
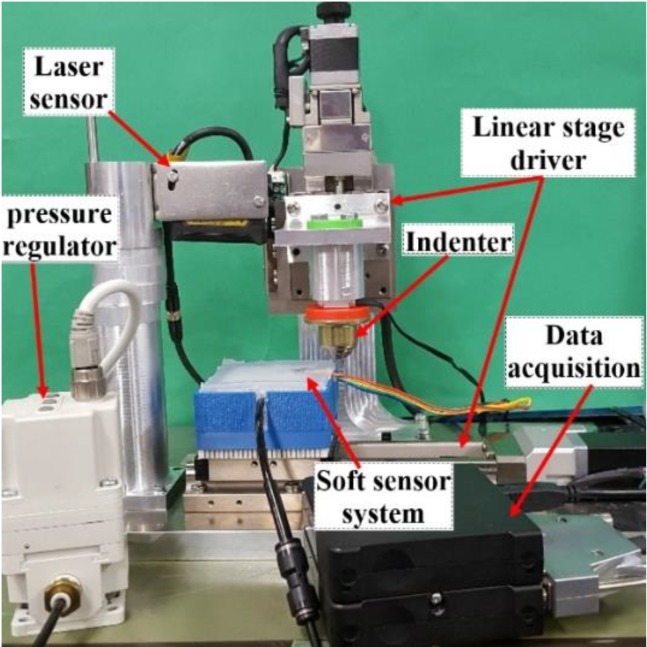
Experimental setup: sensing system is set on horizontal linear stage. Indenter is fixed to vertical linear stage. Skin is inflated by air regulator. Laser sensor measures inflated skin’s height. Data from accelerometer are obtained through data acquisition system before being sent to computer.

**Figure 8 sensors-19-02036-f008:**
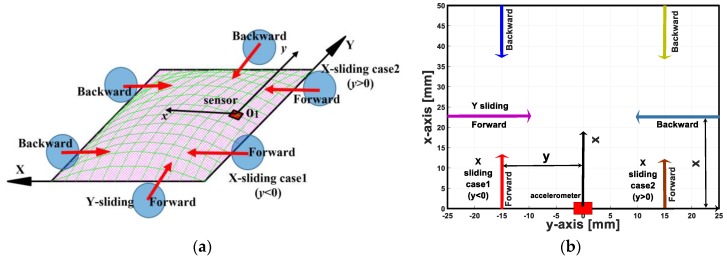
Schematic drawing illustrates sliding cases were conducted in experiments. (**a**) Definition of six sliding directions with both X, Y axes. (**b**) Sliding movements with various contact positions: different values of *y*, *x* for X, Y-sliding.

**Figure 9 sensors-19-02036-f009:**
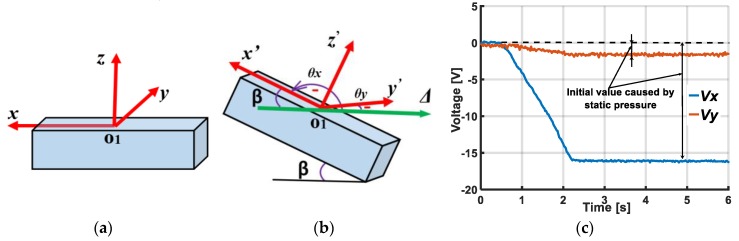
Posture of accelerometer changed and its output voltages under static pressurization. (**a**) Initial posture of accelerometer without pressurization. (**b**) Posture of accelerometer under static pressurization. (c) Output voltages of accelerometer under static pressurization.

**Figure 10 sensors-19-02036-f010:**
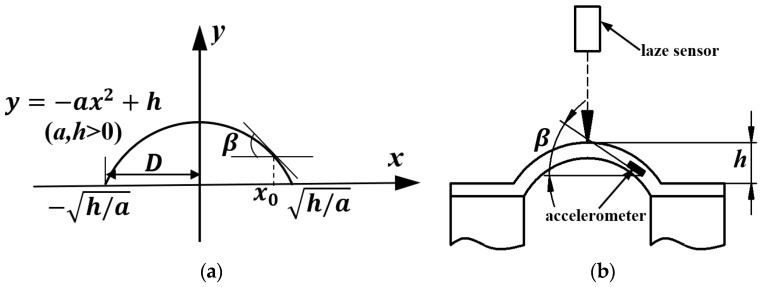
Schematic diagrams to calculate output voltage $V_{x}$ under static pressurization (**a**) Skin’s surface approximated by quadratic curve. (**b**) Schematic drawing of experimental setup for measuring inflated skin’s height.

**Figure 11 sensors-19-02036-f011:**
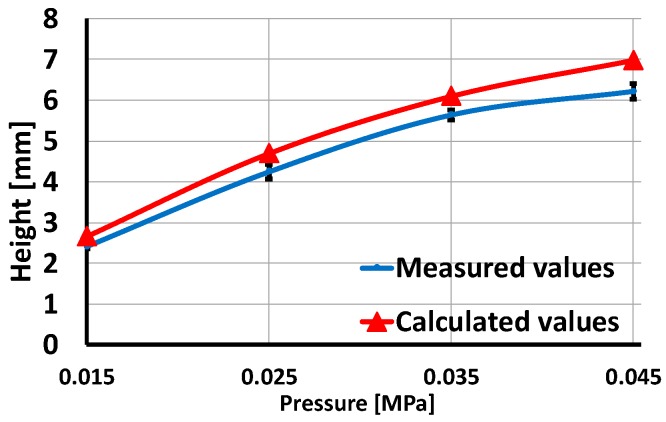
Comparison of calculated and measured values of inflated skin’s height.

**Figure 12 sensors-19-02036-f012:**
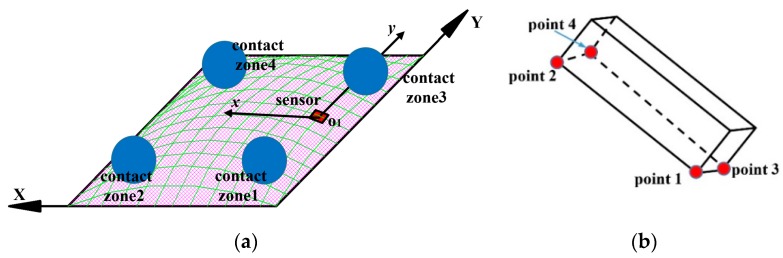
Divided contact zones on skin’s surface and intial posture of accelerometer. (**a**) Depending on location of indenter on skin’s surface, contact area can be categorized into four zones. (**b**) Posture of accelerometer under static pressurization without sliding actions.

**Figure 13 sensors-19-02036-f013:**
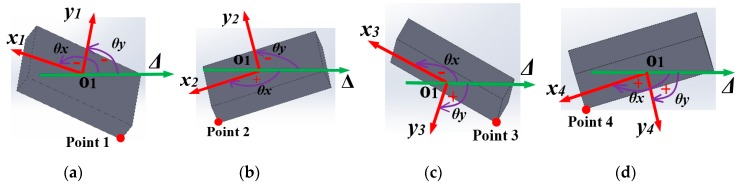
Accelerometer’s posture and its outputs with different contact zones. (**a**) Contact zone 1: Accelerometer is inclined toward point 1 $\theta_{x}<0\to V_{x}<0,\;\theta_{y}<0\to V_{y}<0$. (**b**) Contact zone 2: Accelerometer is inclined toward point 2 $\theta_{x}>0\to V_{x}>0$, $\theta_{y}<0\to V_{y}<0$. (**c**) Contact zone 3: Accelerometer is inclined toward point 3 $\theta_{x}<0\to V_{x}<0,\;\theta_{y}>0\to V_{y}>0$. (**d**) Contact zone 3: Accelerometer is inclined toward point 4 $\theta_{x}>0\to V_{x}>0,\;\theta_{y}>0\to V_{y}>0$.

**Figure 14 sensors-19-02036-f014:**
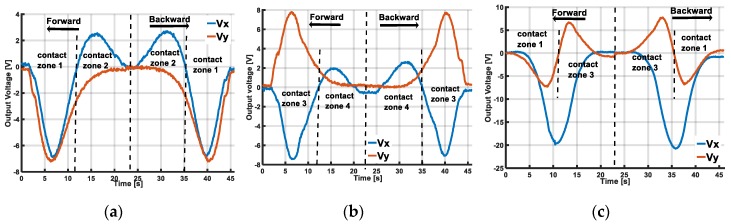
Output voltage $V_{x}$, $V_{y}$ of accelerometer with different sliding cases. $V_{x}$, $V_{y}$ values depend on contact zone between indenter and skin’s surface. Based on changed behaviour of $V_{x}$, $V_{y}$, our sensing system can temporally detect sliding directions: forward and backward in both X- and Y-sliding cases. (**a**) X-sliding case 1. (**b**) X-sliding case 2. (**c**) Y-sliding.

**Figure 15 sensors-19-02036-f015:**
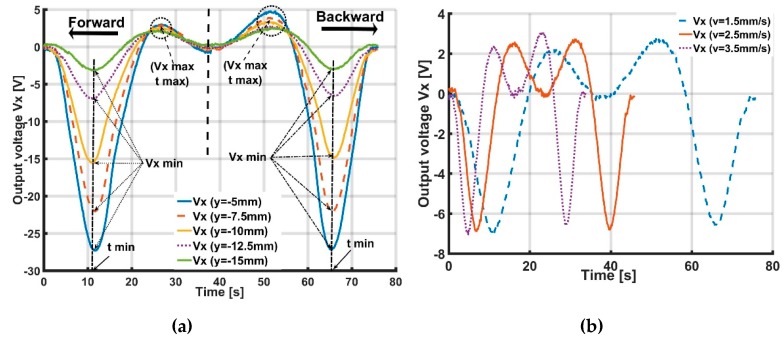
Output voltage $V_{x}$ of accelerometer with X-sliding case 1 when distance y increases from 5 to 15 mm. Value of $V_{x}min$, $V_{x}max$ only depends on distance y and $t_{min},t_{max}$ depends on magnitude of velocity v. Based on accelerometer’s output voltage, distance y from accelerometer to sliding line and magnitude of velocity v can be estimated. (**a**) Output voltage $V_{x}$ with different distances y (mm). (**b**) Output voltage $V_{x}$ with different magnitudes of velocity v (mm/s).

**Figure 16 sensors-19-02036-f016:**
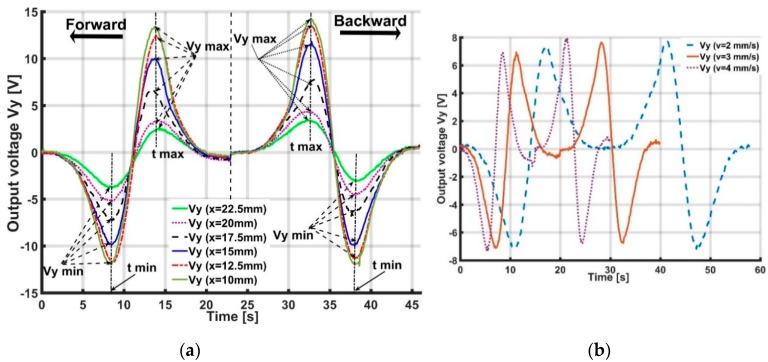
Output voltage $V_{y}$ of accelerometer with Y-sliding when distance x increases from 10 to 22.5 mm. Values of $V_{y}min$, $V_{y}max$ only depend on distance x and $t_{min},t_{max}$ depends on magnitude of velocity v. Based on accelerometer’s output voltage, distance x from accelerometer to sliding line and magnitude of velocity v can be estimated. (**a**) Output voltage $V_{y}\;\left(V\right)$ with different distances x (mm). (**b**) Output voltage $V_{y}\;\left(V\right)$ with different velocities v(mm/s).

**Figure 17 sensors-19-02036-f017:**
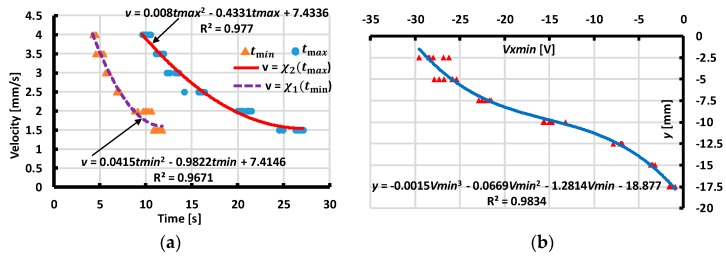
Interpolating velocity v and distance y for case of X-sliding forward. (**a**) Interpolating velocity v from $t_{min},t_{max}$. (**b**) Interpolating distance y from output voltage $V_{x}min$.

**Figure 18 sensors-19-02036-f018:**
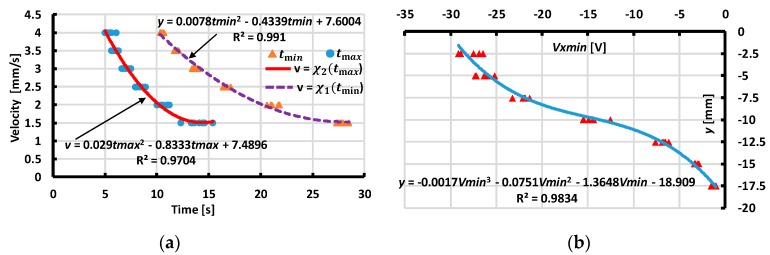
Interpolating velocity v and distance y for case of X-sliding backward (**a**) Interpolating velocity v from $t_{min},t_{max}$. (**b**) Interpolating distance y from output voltage $V_{x}min$.

**Figure 19 sensors-19-02036-f019:**
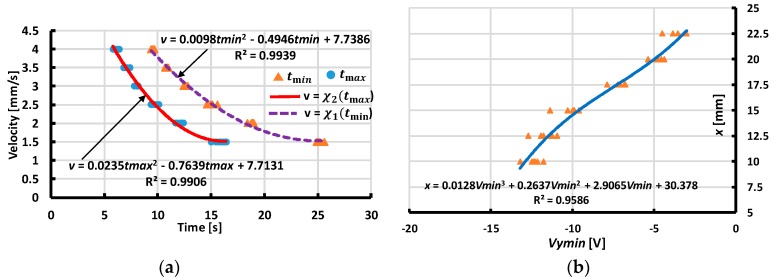
Interpolating velocity v and distance x for case of Y-sliding forward. (**a**) Interpolating velocity v from $t_{min},t_{max}$. (**b**) Interpolating distance x from output voltage $V_{y}min$.

**Figure 20 sensors-19-02036-f020:**
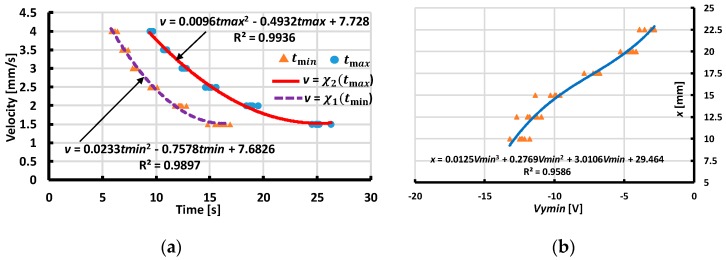
Interpolating velocity v (mm/s) and distance x (mm) for case of Y-sliding backward. (**a**) Interpolating velocity v from $t_{min},t_{max}$. (**b**) Interpolating distance x from output voltage $V_{y}min$.

**Figure 21 sensors-19-02036-f021:**
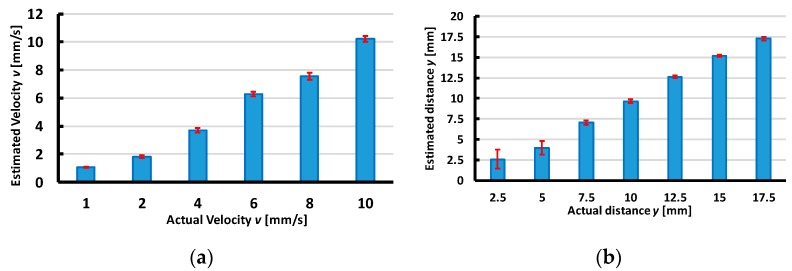
Calculated results compare to actual values with X-sliding case. (**a**) Comparison of calculated and actual velocity v (mm/s). (**b**) Comparison of calculated and actual distance y (mm).

**Figure 22 sensors-19-02036-f022:**
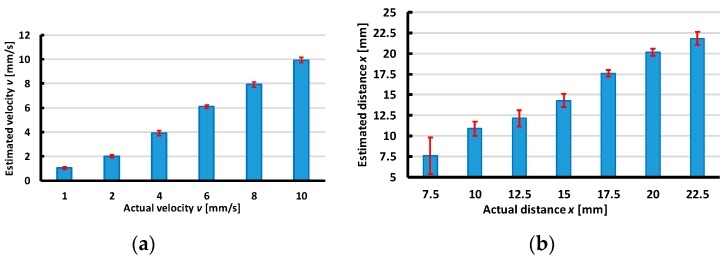
Calculated results compared to actual values with Y-sliding case. (**a**) Comparison of calculated and actual velocity. (**b**) Comparison of calculated and actual distance *x* (mm).

**Figure 23 sensors-19-02036-f023:**
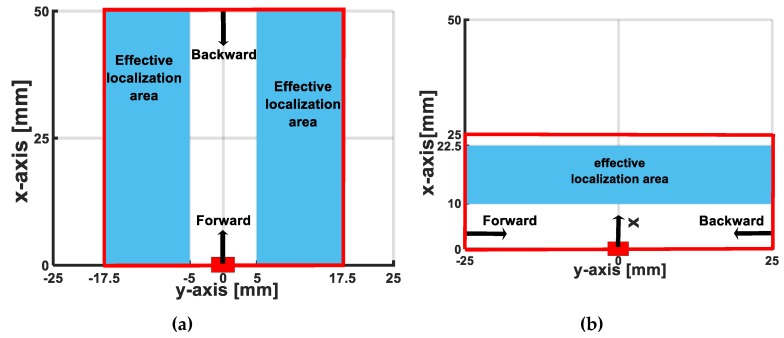
Schematic illustration of detectable area of tactile sensing system. In detectable area (inside red lines), sensing system can detect sliding direction and estimate velocity where maximum errors are less than 8%, and in effective localization area (blue areas), distance from accelerometer to sliding objects can be estimated where maximum errors are less than 15%. (**a**) X-sliding case. (**b**) Y-sliding case.

**Figure 24 sensors-19-02036-f024:**
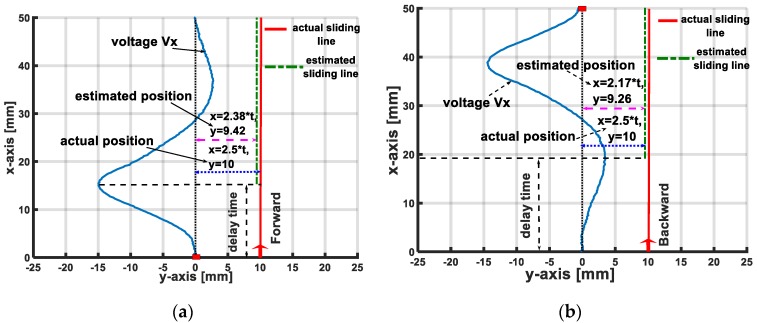
Comparison of actual sliding position and estimated position for X-sliding cases. Actual sliding trajectory is red line, after delay time (when accelerometer’s output voltage reaches extreme values), sensing system can localize sliding object’s positions with errors less than 15% (localized sliding trajectory is blue line). (**a**) X-sliding case: forward direction. (**b**) X-sliding case: backward direction.

**Figure 25 sensors-19-02036-f025:**
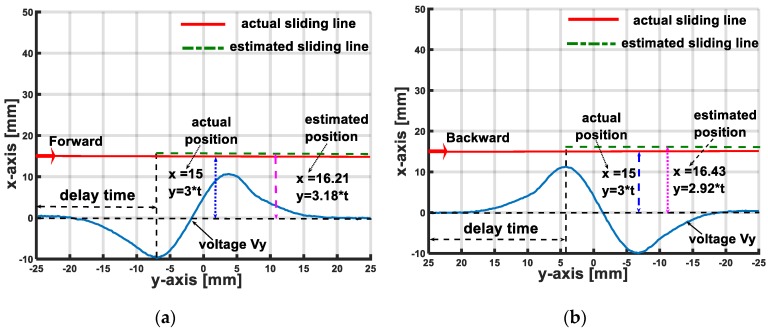
Comparison of actual sliding position and estimated position for Y-sliding cases. (**a**) Y-sliding case: forward direction. (**b**) Y-sliding case: backward direction.

**Figure 26 sensors-19-02036-f026:**
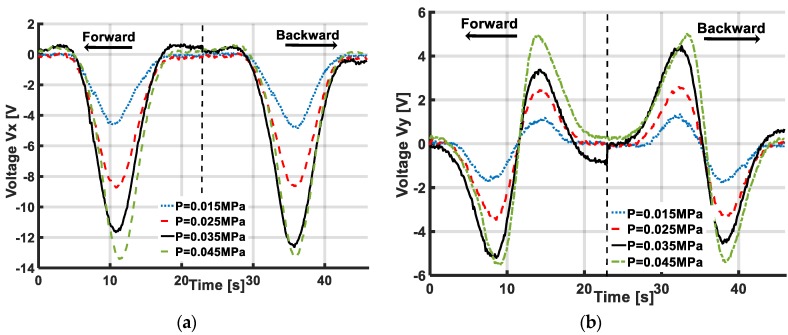
Accelerometer’s output voltage with different pressurizations in Y-sliding case. With higher pressure value, sensing system is more sensitive to sliding action. Nonetheless, high pressurization is relatively ineffective. (**a**) Output voltage $V_{x}\;\left(V\right)$ with different pressurizations. (**b**) Output voltage $V_{y}\;\left(V\right)$ with different pressurizations.

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
