# Peer review of "Localization of Sliding Movements Using Soft Tactile Sensing Systems with Three-axis Accelerometers"

_sensors, 2019, doi:10.3390/s19092036_

Round 1
Reviewer 1 Report
This work presents the development of a tactile sensor based on the tilt of an accelerometer on an inflacted elastomeric substrate. I can't find another example of where accelerometers have been used in quite this way, but there are several examples of accelerators being used to detect the slope of flexure systems and the difference needs to be made clear to confirm novelty. It is also not clear as to what sort of application this technique will be useful for, given its size. The analysis and experimentation is reasonable.
A few comments
There are many tactile sensors and uses of accelerometers for equivalent tasks in the literature. The review in the introduction is not comprehensive enough and comparable techniques need to be critiqued more.
The application of this technology needs to be given more thought. Provide information as to how this technology could be implemented and its relative performance compared to equivalent technologies.
The robustness of the system has to be confirmed. Can it be used to grip anything, or is it just a scale?
Quadratic approximation in eq. 11 needs better justifocation, especially after deformation upon application of a load.
References 23-26 are not formatted correctly
Author Response
We would like to thank Reviewer1 for worthy comments. We have tried our best to take all in account and revise the manuscript to meet requirements from the reviewer. Detailed response, edited parts in the revised manuscript are shown as follows and our revised manuscript as in the attached file.
This work presents the development of a tactile sensor based on the tilt of an accelerometer on an inflated elastomeric substrate. I can't find another example of where accelerometers have been used in quite this way, but there are several examples of accelerators being used to detect the slope of flexure systems and the difference needs to be made clear to confirm novelty. It is also not clear as to what sort of application this technique will be useful for, given its size. The analysis and experimentation is reasonable.
Q1: There are many tactile sensors and uses of accelerometers for equivalent tasks in the literature. The review in the introduction is not comprehensive enough and comparable techniques need to be critiqued more.
Answer 1:
We appreciated this comment and seriously considered its importance in making the introduction section of our manuscript completed. To respond to this comment, we have referred many other tactile sensors based on the accelerometers to confirm the novelty of our paper. We added more content to the introduction section of our revised manuscript (colored text in page 2 in the revised manuscript), also shown as follows for your quick look:
Several studies utilized accelerometers for fabrication of tactile sensors [19-21]. A previous work in [19] used a miniature accelerometer that was attached to a human finger to measure the acceleration at the radial skin to estimate the surface undulation. It was also attached to several pneumatic grippers to classify the hardness of different cylinders, to estimate the pneumatic pressure, and to assess the firmness of eggplants and mangoes [20]. Authors in [21] proposed a bio-mimetic fingertip on which they embedded three commercial accelerometers and force sensors to detect force and vibration modalities. Most accelerometer applications in tactile sensors focus on estimating physical quantities, such as hardness and firmness and using measured vibrations for surface identification. In terms of detecting and localization sliding movements, there is a lack of interest in accelerometer applications in tactile sensing systems.
Q2: The application of this technology needs to be given more thought. Provide information as to how this technology could be implemented and its relative performance compared to equivalent technologies. The robustness of the system has to be confirmed. Can it be used to grip anything, or is it just a scale?
We would like to thank this comment. It is meaningful to clarify the possible applications of our tactile sensing system. To discuss this issue, we added new section “5.4. Limitations and applicability of the proposed sensing system” in our revised manuscript. This section includes the information of sensing system’s applicability. It is shown as follows for your quickly look:
Our sensing system is anticipated for applications in soft robotics. For instance, it could be attached to a soft robotics hand’s palm or other body parts of robots that frequently make contact with sliding objects. Normally, without pressurization, a soft skin surface plays the role of robotic skin and is inflated for detecting and localization sliding tasks, which are crucial for robots to make subsequent decisions. Because the pressurization for activating the inflated skin is small (maximum, 0.035 MPa), a sensing system can be activated with a small pump and a simple pneumatic controller without affecting the portability of the whole device. In addition, our sensor system is a promising device for detecting sliding objects and localizing their position in an environment with limited vision. In this paper we avoided applying our sensing system for specific applications. Instead, we set a prototype design, fabrication, and experiment with analysis to validate its ability. Thus, other researchers can exploit it to design specifically purposed structures.
Our sensing system can detect and localize a sliding movement, including sliding directions, velocity and object’s locations. This design is scalable. If the system could be attached to the gripper or robotics hand, it can be applied to grip objects. The integration of tactile sensing and gripping tasks need a further investigation. This issue will to be tackled in our future work.
Q3: Quadratic approximation in eq. 11 needs better justification, especially after deformation upon application of a load.
Answer 3:
In section 5.1, we estimate the accelerometer’s output voltage under static pressure at the no-sliding cases. Thus, the quadratic approximation in Eq. (11) (Eq. (9) in our revised manuscript) was given to estimate the inflated skin surface under only the pressurization’s activation without the application of load or sliding motion.
The computed values of the inflated skin’s height from Eq. (11) (Eq. (9) in our revised manuscript) is compared to the measured ones using a laser sensor. This comparison verifies that in the case of only static pressurization, based on the accelerometer’s output voltage we can accurately estimate the accelerometer’s posture as well as the inflated skin’s shape. The error of this comparison is less than 12%, thus the quadratic approximation in Eq. (11) is considered acceptable.
Q4: References 23-26 are not formatted correctly
Answer 4:
Thank you so much for pointing out this issue. In our revised manuscript, it was corrected.

Reviewer 2 Report
The paper presents a kind of tactile sensor based on an accelerometer embedded in an elastic layer that is placed on a pressurized elastic chamber.
A contact on the surface of the device causes a mechanical deformation and it is detected by the accelerometer acting as an inclinometer.
The idea and work behind the paper do not lack of merit. Nevertheless, the way it is presented is quite confusing, I think there is much room to improve the style of the paper and make it clearer. Moreover, there are fundamental issues and limitations of the idea that could be solved in improved versions, but must be discussed in the paper.
First, I do not see (please explain it) why you do not present the idea in a more straightforward way. I mean that you infer the position of the object from the accelerometer readings, once you have characterized the mechanical behavior of the device to stablish the relation between the accelerometer readings and the location of the object. Once you know the location, you can obtain easily the velocity. I do not see the point in highlighting the ability to detect sliding and velocity first, and then location. From my point of view (I could have misunderstood something), once you have the accelerometer readings and the relation with the location of the object on the surface of the device, you can infer readily the velocity.
Second, you suppose that the surface is flat and neglect the reading of Vz. This should be discussed because the surface shape can change with the pressure in the chamber.
Third, you do not mention the force at the contact point. It is clear that you obtain the same reading for two pairs of forces, locations. There is a clear inversion problem here. You should discuss it.
Another limitation related to the principle behind the device is that the sensor readings change if the whole device changes its posture. This can be solved with a second inclinometer and some processing, but it also should be mention in my opinion.
The role of other accelerations, for instance those due to friction between the object and the surface of the device should also be mentioned. Filtering of the use or a more complex IMU could be ways to reduce their effects.
It is not clear to me why the voltages are larger when the pressure is higher. There must be a higher deformation of the surface of the device to obtain higher readings from the accelerometer. Please clarify this.
Regarding the presentation, the English is generally good, though I have detected some little mistakes such as:
Line 41: exsamples
Line 231: ??, ?? is the angle between instead of ??, ?? ARE the angleS between…
The use of an accelerometer as inclinometer is well known, I would not highlight this as a contribution of the paper, as it seems you do in the abstract and the body of the paper. Please look for a way to explain the content in a simpler and more straightforward way and use only the equations that are really needed to follow the paper (I do not see the purpose of showing equation (6), for instance, and I think equations from (1) to (6) could be summarized, the use of the accelerometer as inclinometer is already known, as I said).
As a final comment, the comparison with the web of a spider is beautiful, you can keep it if you like, but the spider actually detects vibrations, I mean you can compare it with your accelerometer detecting accelerations caused by vibrations but not with the gravity.
Author Response
We would like to thank Reviewer 2 for worthy comments with insightful review. We have tried our best to take all in account and revise the manuscript to meet requirements from the Reviewer. Detailed response, edited parts in the revised manuscript are shown as follows and our revised manuscript as in the attached file.
Q1:
The paper presents a kind of tactile sensor based on an accelerometer embedded in an elastic layer that is placed on a pressurized elastic chamber.
A contact on the surface of the device causes a mechanical deformation and it is detected by the accelerometer acting as an inclinometer.
The idea and work behind the paper do not lack of merit. Nevertheless, the way it is presented is quite confusing, I think there is much room to improve the style of the paper and make it clearer. Moreover, there are fundamental issues and limitations of the idea that could be solved in improved versions, but must be discussed in the paper.
First, I do not see (please explain it) why you do not present the idea in a more straightforward way. I mean that you infer the position of the object from the accelerometer readings, once you have characterized the mechanical behavior of the device to stablish the relation between the accelerometer readings and the location of the object. Once you know the location, you can obtain easily the velocity. I do not see the point in highlighting the ability to detect sliding and velocity first, and then location. From my point of view (I could have misunderstood something), once you have the accelerometer readings and the relation with the location of the object on the surface of the device, you can infer readily the velocity.
Answer1:
- In our revised manuscript the fundamental issues and limitations of the sensing system is discussed in Section 5. 4 (detailed response as in the answers of your questions 2 to 4).
- In this paper, we focus on the localization of sliding movements in two X- and Y- directions with assumption that at first a sliding object slightly contacts the soft skin outside the inflated area and then slides over the inflated skin at a constant velocity without changing the initial contact depth. The localization of the sliding object can be characterized by the sliding directions and its position coordinate (x, y). Thus, for localization of sliding movement, first the sensing system must detect the sliding directions then estimate the object’s position (x, y) (Figure 1(b) in revised manuscript).
For X-sliding case, y is the distance from the sliding line to the accelerometer, its value is constant with each sliding line and can be calculated from the accelerometer’s output. However, x is function of time and the velocity’s magnitude. If we infer the value of x from the accelerometer’s output, required calculation is so complicated due to the number of variable x is very large. Thus, instead directly estimate the position of the object, we estimate the magnitude of velocity then infer the position. Similar explanation for Y-sliding case.
To clarify this issue in Section 2.1 of our revised manuscript we added more content to present the idea in more straightforward way as your recommendation. (line 74-84 in page 3 of the revised manuscript), also shown as follows for your quickly look:
In this paper, we focus on the localization of sliding movements in two X- and Y- directions with different distances from a sliding line to the accelerometer (Figure 1(b)). For simplicity, we suppose that first a sliding object slightly contacts the soft skin (outside the inflated area) and then slides over the inflated area at a constant velocity without changing the initial contact depth. The localization of the sliding object can be characterized by the sliding directions and its position coordinate (x, y). For the X-sliding case, y is the distance from the sliding line to the accelerometer and x = vt. Whereas for the Y-sliding case, x is the distance from the sliding line to the accelerometer and y = vt. Here v is the magnitude of sliding velocity. Thus, for localization, first we demonstrate that the sensing system can detect the sliding motion and its directions. Based on the accelerometer output, the sliding velocity’s magnitudes and the distances from the sliding line to the accelerometer are estimated. Then the sliding object can be localized.
In addition, we also added more content in section 5.2 (Line 256-258 in page 10, line 306 -309 in page 12, line 328-331 in page 12, and line 332-335 page 13) to highlight the ability to detect sliding and velocity first, and then location. These revised contents are shown as followed for your quickly look:
Line 256-258 in page 10
To localize the sliding movements, first the sensing system needs to recognize the motion’s directions. In this section, we propose dynamic analysis based on the accelerometer’s output for detecting the sliding directions.
Line 306 -309 in page 12
We presented the idea of sliding movement’s localization in Section 2.1 and in Section 5.2.1 demonstrated that the sensing system can temporally detect a sliding motion and its directions. Based on the accelerometer’s output voltage, in this section we confirm that the sliding object can be localized through the distance from the sliding line to the accelerometer and the magnitude of the sliding velocity.
Line 328-331 in page 12
Figure 15. Output voltage Vx of accelerometer with X-sliding case 1 when distance y increases from 5 to 15 mm. Value of Vxmin, Vxmax only depends on distance y and tmin, tmax depends on magnitude of velocity v. Based on accelerometer’s output voltage, distance y from accelerometer to sliding line and magnitude of velocity v can be estimated.
Line 332-335 page 13
Figure 16. Output voltage Vy of accelerometer with Y-sliding when distance x increases from 10 to 22.5 mm. Values of Vymin, Vymax only depend on distance x and tmin, tmax depends on magnitude of velocity v. Based on accelerometer’s output voltage, distance x from accelerometer to sliding line and magnitude of velocity v can be estimated.
Q2.
Second, you suppose that the surface is flat and neglect the reading of Vz. This should be discussed because the surface shape can change with the pressure in the chamber.
Answer 2:
In our paper, we neither used the assumption of a flat surface, nor neglected the output voltage Vz. Note that without pressurization the skin’s surface is flat and under the activation of pressure, it is inflated, making the sensitivity of the sensing system to the sliding action on its surface. The shape of inflated skin surface with the static pressure was estimated in Section 5.1.
In addition, because in this paper we focused on the localization of sliding movements in two X- and Y-directions, thus we used the output voltage Vx, Vy. Vz is useful for estimation of 3D position of objects. We now only consider two-dimensional localization, so we do not need to use Vz.
Q3:
Third, you do not mention the force at the contact point. It is clear that you obtain the same reading for two pairs of forces, locations. There is a clear inversion problem here. You should discuss it.
Another limitation related to the principle behind the device is that the sensor readings change if the whole device changes its posture. This can be solved with a second inclinometer and some processing, but it also should be mention in my opinion.
The role of other accelerations, for instance those due to friction between the object and the surface of the device should also be mentioned. Filtering of the use or a more complex IMU could be ways to reduce their effects.
Answer 3:
We greatly appreciate this insightful comment. We seriously considered its importance in making our manuscripts completed. These limitations you pointed out in our manuscripts are very accurate. In our revised manuscript, we added a new section 5.4. “Limitations and applicability of sensing system”. In this section we have discussed the limitations of sensing system and potential solutions in next our work. Our discussion in the revised manuscript is shown as followings for your quickly look:
The accelerometer’s output depends on the contact location and the contact force. We assume that first the sliding object slightly contacts the soft skin outside the inflated area and slides without changing the initial contact depth. Thus, under specific pressurization, the contact force at each contact point is identical for all sliding movements. We can ignore the effect of the contact force on the accelerometer’s output voltage. In the case of changing the contact depth during sliding movements, the effect of the force at each contact point must be considered. To solve this issue, in the next step we will propose a finite element model to estimate the deformation of the inflated soft skin’s surface under various contact forces. This model could also be used to locate the indentations on soft skin and estimate the normal contact force. In addition, the effect of acceleration due to the friction force between the sliding object and the skin’s surface was not mentioned in this paper. This effect could be reduced by filtering, such as with a Kalman filter.
A sensing system can be used for specific devices and an accelerometer’s output may change if the entire device changes its posture. Thus, this effect must be investigated and eliminated. To solve this issue, it may require an algorithm that calculates the accelerometer’s output based on the whole device’s posture. These calculated values will be subtracted from the accelerometer’s acquired output for sensing.
Another current limitation of our sensing system is that it can detect and localize sliding motions with two X- and Y- directions because we only deploy the accelerometer as a sensing element. In the future we will use more accelerometers, each of which will be placed at a suitable initial posture and position. Based on the output responses of all the accelerometers, we will build a method using machine learning to localize the sliding motions with diagonal directions. To accomplish complicated tasks, a sensing system must be trained with training data in various conditions of sliding motions. After training, the acquired program can be used as the sensing system’s brain that should be attached to the main processing system. The processing system can help a sensing system be localized temporally with improved revolutions. We must tackle this critical issue in the future to fulfill this sensing system. In addition, we will use another accelerometer with a flexible circuit to replace the current accelerometer with a solid circuit board. For example, the previously proposed flexible one [21] would be easier to embed in soft skin.
Q4:
It is not clear to me why the voltages are larger when the pressure is higher. There must be a higher deformation of the surface of the device to obtain higher readings from the accelerometer. Please clarify this.
Answer 4:
Thank you for your comment. We agree with your opinion that when the pressure is higher, the deformation of the skin’s surface is increased, thus the accelerometer’s output increases. As shown in Figure 11 in Section 5.1, with higher pressurization, the skin’s height increases, leading to higher deformation of the skin’s surface under identical sliding motion. Resulting in increased output voltage of the accelerometer. This issue is also validated by simulation results in Section 3 in our manuscript.
To clarify this issue, we added more content in Section 5.3 in our revised manuscript (red text in line 393 to 398 and line 406 to 407), shown as follows for your quickly look:
Line 393 to 398
In this proposed tactile sensing system, inflated soft skin is generated by the activation of pressurization. Pressurization plays the actuation role that creates the continued change in the accelerometer’s position under the object’s sliding action. Figure 11 in Section 5.1 shows that with higher pressurization, the inflated skin’s height increases, leading to higher deformation of the skin’s surface under identical sliding motions. Resulting in increased output voltage of the accelerometer.
Line 406 to 407
Thus, by confirming the simulation and experimental results, we conclude that with higher pressure values, the sensing system’s sensitivity increases with sliding motions.
Q5:
Regarding the presentation, the English is generally good, though I have detected some little mistakes such as:
Line 41: exsamples
Line 231: ??, ?? is the angle between instead of ??, ?? ARE the angleS between…
Answer 5:
Thank you so much for your pointing out our mistakes, these mistakes were edited in our revised manuscript.
Q6:
The use of an accelerometer as inclinometer is well known, I would not highlight this as a contribution of the paper, as it seems you do in the abstract and the body of the paper. Please look for a way to explain the content in a simpler and more straightforward way and use only the equations that are really needed to follow the paper (I do not see the purpose of showing equation (6), for instance, and I think equations from (1) to (6) could be summarized, the use of the accelerometer as inclinometer is already known, as I said).
Answer 6:
We totally agree with your opinion that the use of an accelerometer as inclinometer is well known. However, using the accelerometer as inclinometer to detect and localize sliding motion is not easy to understand for most readers. Thus, in this paper, we attempt to set the simplest way to explain our sensing system’s operation principle. Besides, the explanation in Section 2.3 will help the reader easy to follow our calculations and analysis in next Sections.
However, we agree with the Reviewer that the explanation of the sensing system’s principle based on the accelerometer is not one of the main contributions of our paper. In our revision, we edited the abstract and Section 2.3 of the paper. Equations 4 and 6 in Section 2.3 of our previous manuscript were removed. The abstract was also revised, also shown as follow for your quick look:
“First, we present the idea, design, fabrication process, and the operation principle of our proposed sensor. Next, we created a numerical simulation model to investigate the dynamic changes of the accelerometer’s posture under sliding actions. Finally, experiments were conducted with various sliding conditions. By confirming the numerical simulation, dynamic analysis, and experimental results, we acknowledge that the sensor system can detect the sliding movements, including the sliding directions, velocity, and localization of an object. We also point out the role of pressurization in the sensing system’s sensitivity under sliding movements, implying the ideal pressurization for it. We also discuss its limitations and applicability.”
Q7:
As a final comment, the comparison with the web of a spider is beautiful, you can keep it if you like, but the spider actually detects vibrations, I mean you can compare it with your accelerometer detecting accelerations caused by vibrations but not with the gravity.
Answer 7:
Thank you for your recommendation. We agree with your comment that in this paper we don’t use vibrations to detect sliding objects. However, the main idea of our paper is detecting and localization sliding object on large contact surface. Thus, we create the inflated skin surface under activation of pressurization, this surface spreads the deformation from the contact’s position to the accelerometer’s location, which mimics the role of a spider’s web in a vibration’s transmission. To avoid misunderstanding of reader, we added more content to the Section 2.1 of our revised manuscript (colored text in line 62 to 67, page 3), also shown as follow for you quick look:
Under pressurization, a curved surface was generated, and when an object contacts and slides on it, the deformation at the contact’s position is conveyed to the accelerometer’s location that provides information about the acceleration of the three axes (x, y and z axes). This information is sufficient for detecting the sliding directions and localizing the sliding movement. In our sensing system, the inflated skin’s surface spreads the deformation, which mimics the role of a spider’s web in a vibration’s transmission.

Round 2
Reviewer 2 Report
Thank for addressing my suggestions.
Nevertheless, I do not understand the sentence "If we infer the value of x from the accelerometer’s output,
required calculation is so complicated due to the number of variable x
is very large" in your response to my first question. Please take into account that your paper wil be read and cited if it is understood. I guess you mean that a real time computation of the location of x is not possible for the velocities you are considering...